

# Foraging loads of red wood ants: *Formica aquilonia* (Hymenoptera: Formicidae) in relation to tree characteristics and stand age

Heloise Gibb[1,2], Jon Andersson[1] and Therese Johansson[1]

[1] Department of Wildlife, Fish and Environmental Sciences, Swedish University of Agricultural Sciences, Umea, Sweden

[2] Department of Ecology, Environment and Evolution, La Trobe University, Melbourne, Australia

## ABSTRACT

**Background.** Foraging efficiency is critical in determining the success of organisms and may be affected by a range of factors, including resource distance and quality. For social insects such as ants, outcomes must be considered at the level of both the individual and the colony. It is important to understand whether anthropogenic disturbances, such as forestry, affect foraging loads, independent of effects on the quality and distribution of resources. We asked if ants harvest greater loads from more distant and higher quality resources, how individual efforts scale to the colony level, and whether worker loads are affected by stand age.

**Methods.** First, we performed a fine-scale study examining the effect of distance and resource quality (tree diameter and species) on harvesting of honeydew by red wood ants, *Formica aquilonia*, in terms of crop load per worker ant and numbers of workers walking up and down each tree (ant activity) (study 1). Second, we modelled what the combination of load and worker number responses meant for colony-level foraging loads. Third, at a larger scale, we asked whether the relationship between worker load and resource quality and distance depended on stand age (study 2).

**Results.** Study 1 revealed that seventy percent of ants descending trees carried honeydew, and the percentage of workers that were honeydew harvesters was not related to tree species or diameter, but increased weakly with distance. Distance positively affected load mass in both studies 1 and 2, while diameter had weak negative effects on load. Relationships between load and distance and diameter did not differ among stands of different ages. Our model showed that colony-level loads declined much more rapidly with distance for small diameter than large diameter trees.

**Discussion.** We suggest that a negative relationship between diameter and honeydew load detected in study 1 might be a result of crowding on large diameter trees close to nests, while the increase in honeydew load with distance may result from resource depletion close to nests. At the colony level, our model suggests that very little honeydew was harvested from more distant trees if they were small, but that more distant larger trees continued to contribute substantially to colony harvest. Although forestry alters the activity and foraging success of red wood ants, study 2 showed that it does not alter the fundamental rules determining the allocation of foraging effort.

Corresponding author
Heloise Gibb, h.gibb@latrobe.edu.au

## INTRODUCTION

Foraging efficiency is critical in determining the success of organisms. Our understanding of foraging efficiency has been shaped by theories of optimal foraging, whereby organisms are predicted to forage to maximise their energy intake per unit time (*MacArthur & Pianka, 1966*). Where animals return to a central place (e.g., a nest), central place foraging theory, in its original formulation, predicted that more valuable resources or larger loads would be harvested at greater distances from the central place (*Orians & Pearson, 1979*). This is because the energy expenditure of foragers is proportional to the distance travelled. Positive load-distance relationships have previously been observed in a range of taxa that act as central place foragers (*Giraldeau & Kramer, 1982*; *Mellgren, Misasi & Brown, 1984*; *Kacelnik, Houston & Schmidhempel, 1986*; *Kaspari, 1991*). However, more recent models of central place foraging suggest that observed positive load-distance relationships may represent a special case, where costs associated with greater travel time and resource loads, including missed opportunity, metabolic and predation risk costs, are negligible (*Olsson, Brown & Helf, 2008*). This may explain why some studies have failed to detect a positive load-distance relationship (e.g.,*Wetterer, 1991*). Alternatively, positive load-distance relationships might also occur where resource quality or quantity increases with distance from nest (*Olsson, Brown & Helf, 2008*). This pattern might be expected if foraging activity results in resource depletion, a phenomenon that is particularly likely to be important for colonial organisms, such as ants.

As central place foragers, ants commonly show positive load-distance relationships in empirical studies. This pattern occurs frequently where workers control load size (leaf-cutting, *Roces, 1990*; e.g., liquid food harvesting, *Bonser et al., 1998*; *Wright, Bonser & Chukwu, 2000*), but is less prevalent where resource size is fixed (e.g., seeds, *Rissing & Pollock, 1984*; *Holder & Polis, 1987*). While this appears inconsistent with recent models, the increase in metabolic costs and predation risk with distance may be negligible for some species: metabolic costs do not increase substantially with foraging load for ants (*Nielsen, Jensen & Holmjensen, 1982*), energetic rewards are orders of magnitude higher than worker energetic expenditure (*Baroni-Urbani & Nielsen, 1990*), and increases in foraging speed with resource distance might minimise additional exposure to predation (*Torres-Contreras & Vasquez, 2004*). Further, the higher density of foraging workers close to colonies (*De Vita, 1979*; *Savolainen & Vepsäläinen, 1988*) suggests that resource depletion (exploitation competition) or intra-colonial interference competition might drive individuals to forage at greater distances from the nest, where more desirable load sizes might be obtained (*Wright, Bonser & Chukwu, 2000*; *Grüter et al., 2012*). Positive load-distance relationships for social insects, such as ants, might therefore result from increases in resource quality and quantity with distance. However, positive load-distance relationships have also been observed in laboratory-based studies, where resource quality and quantity is tightly controlled (e.g., *Bonser et al., 1998*).

Although most studies of central place foraging in ants focus on individual workers (e.g., *Holder & Polis, 1987*; *Bonser et al., 1998*; *Wright, Bonser & Chukwu, 2000*), the aim of workers should be to maximise the resources harvested by the colony as a whole, by altering

both individual loads and the number of foragers active (*Burd & Howard, 2005*). This is because selection on eusocial organisms is expected to operate most strongly at the level of the colony as workers of most species are sterile (*Bourke & Franks, 1995*). This could occur through recruitment or other cues based on encounter densities and through size-dependent foraging behaviour in polymorphic species. Even in only moderately polymorphic or monomorphic species, larger ants travel greater distances to forage and they are more efficient, i.e., they are able to carry a greater load relative to their body weight (*Herbers & Cunningham, 1983*; *Rosengren & Sundström, 1987*; *McIver, 1991*; *McIver & Loomis, 1993*; *Burd, 1995*; *Wright, Bonser & Chukwu, 2000*) (but see *Rissing & Pollock, 1984*). In addition to distance, the harvesting efficiency of ants is affected by a range of other factors, including resource quality, such as sucrose concentration, carbohydrate:protein ratio, resource quantity, temperature or crowding (*Dreisig, 1988*; *Bonser et al., 1998*; *Cerda, Retana & Cros, 1998*; *Detrain et al., 2000*; *Kay, 2002*; *Segev et al., 2014*).

Habitat structure plays a key role in determining the success of species (e.g., *Petren & Case, 1998*; *Stephens et al., 2004*; *Cushman, 2006*), particularly ants (*Lassau & Hochuli, 2004*; *Sorvari & Hakkarainen, 2004*; *Sarty, Abbott & Lester, 2006*; *Gibb & Parr, 2010*). For example, larger ants are more successful competitors in structurally simple habitats, probably because they are faster to discover and exploit resources (*Gibb & Parr, 2010*). Anthropogenic disturbances, including urbanisation, agriculture and forestry, transform landscapes, significantly altering habitat structure (*Harrison & Bruna, 1999*; *Gibb & Hochuli, 2002*). For example, forestry practices in mid-boreal Sweden have resulted in a disproportionately large area of relatively young and dense stands, with structure differing substantially from old growth stands (*Linder & Östlund, 1998*). Previous studies suggest that stand age (a measure of time since disturbance or successional stage) and the associated structural differences have significant effects on the abundance and behaviour of ants (*Punttila, 1996*; *Sorvari & Hakkarainen, 2004*; *Gibb & Johansson, 2010*) and on resource quality (*Johansson & Gibb, 2012*). However, no previous study has tested whether successional stage affects foraging loads of ants. We might expect differences because later successional stages are associated with increases in resource quality and quantity (*Guariguata & Ostertag, 2001*; *Johansson & Gibb, 2012*), ant density, inter-specific competition and predator richness (*Niemelä, Haila & Punttila, 1996*; *Gibb, 2011*).

The aim of this study was to determine how foraging distance, tree species and diameter (measures of resource quality) and an anthropogenic disturbance (forestry) interact to determine the loads carried by individual worker ants and the consequences for colony resource harvesting when worker activity is accounted for. We used northern red wood ants, *Formica aquilonia* (Yarrow, 1955), in boreal forests to examine, first, whether the crop loads (mass gain) of individual workers change with resource distance and tree species and diameter (study 1). Second, we used activity data from a previous study to model how colony-level mass gain changes with resource distance and diameter for Norway spruce, *Picea abies*. Third, we examined whether the relationship between worker load and tree diameter and distance differed among stands of different ages for spruce (study 2).

## MATERIALS AND METHODS

### Ethics statement

This study complies with the current laws of Sweden. The forestry companies Holmen Skog AB, Sveaskog AB, SCA and Scaninge gave permission to use their land.

### Study sites and species

This study was conducted in boreal forests in northern Sweden between the latitudes of 63.6°N and 64.5°N and longitudes of 19.7°E and 20.7°E. The forest was dominated by Norway spruce, *Picea abies* (70–100%), while birches, *Betula pubescens* and *Betula pendula*, and Scots pine, *Pinus silvestris,* also occurred in significant numbers. The field layer consisted mainly of dwarf shrubs (*Vaccinium spp.*) and soils were moist and of the sandy moraine type. A detailed study of foraging loads was first conducted in a single mature production forest (study 1). The effect of stand age on forager loads was then examined using a further twelve stands (study 2): mature stands ($n = 4$, tree age 80–100 years, non-sapling mean basal diameter (BD) = $30.3 \pm 1.6$ cm, mean height ($H$) = $17.9 \pm 0.6$ m, mean stem density (SD) = $1974 \pm 146$ stems.ha$^{-1}$), middle aged stands ($n = 4$, 30–40 years, BD = $13.8 \pm 0.9$ cm, $H = 8.1 \pm 0.3$ m, SD = $3,923 \pm 471$ stems.ha$^{-1}$) and clear cuts with 5–10 retention trees per ha ($n = 4$, 1–4 years, BD = $4.8 \pm 0.7$ cm, $H = 2.1 \pm 0.3$ m, SD = $942 \pm 50$ stems.ha$^{-1}$). Further details on site characteristics are provided in Table S1 and *Gibb & Johansson, 2010*, Appendix 1. Measures of basal diameter included all trees >1 cm BD. Basal diameter was used in preference to diameter at breast height so that seedlings, which may be shorter than 1.3 m in height, but also provide a food source to ants, could be included. Stands of different ages were geographically interspersed and each study plot supported several nests of the northern red wood ant, *Formica aquilonia*. Mean $\pm$ SE stand separation was $17.5 \pm 1.0$ km.

*F. aquilonia* belongs to the *F. rufa* group, which consists of territorial behaviourally dominant ant species that have been reported to structure ant communities (*Savolainen & Vepsalainen, 1989*; *Gibb, 2011*; *Gibb & Johansson, 2011*). It has polygynous and polydomous colonies throughout its range (*Pamilo, 1982*) and is the most common *F. rufa* group species in the central boreal region of Fennoscandia (*Collingwood, 1979*). In the study area, *F. aquilonia* is commonly observed climbing trees, where it tends the aphids *Cinara pruinosa* and *C. piceicola* (*Johansson & Gibb, 2012*) (aphids identified by R Danielsson, University of Lund and Nils Ericson, Umeå). A previous study showed that honeydew makes up approximately 80% of the diet of *F. aquilonia* in Finland, with the remainder consisting of invertebrate prey (*Domisch et al., 2009*).

### Study 1: Do ants adjust their foraging based on resource quality and distance?

Our aim here was to determine if the quantity of honeydew harvested from a mature tree per forager was affected by resource quality (tree species and basal diameter) or distance from the nest. For this reason, we selected a single mature forest, where variation in stem basal diameter was greater than younger stands. Tree species affects the quality of aphid honeydew (*Douglas, 1993*) and, within tree species, honeydew quality is affected by diameter

through increases in the concentrations of some sugars, including fructose, melezitose and trehalose (*Johansson & Gibb, 2012*), and changes in secondary metabolites (*Whitham, 1978*; *Price, 1991*). Unpublished data from our study sites suggests that the composition of sugars and amino acids differs among honeydew samples collected from *P. abies*, *P. sylvestris* and *Betula spp.* Sampling was performed once at each site on fine days in July 2007 between 9:00 and 17:00. We selected 10 nests of *F. aquilonia* (Mean ± SE nest volume: 0.40 ± 0.16) in mature forest and located two spruce, one birch and one pine tree (basal diameter range: 9.5–48 cm) with high levels of ant activity within 20 m of each nest (thousands of workers active on and around the nest), with clear trails originating from the target nest. The distance from the nest and the basal diameter of each tree was measured and ambient temperatures were recorded for each tree at the time sampling commenced. Nests contained multiple entrance holes, so distances were measured from the centre of the nest. All nests were greater than 60 m from an edge with another habitat type and greater than 50 m from another nest.

We used modified battery-driven vacuum cleaners with aspirators attached to collect 20 ants walking up and 20 ants walking down each of the trees (a total of 40 ants × 10 nests × 4 trees = 1,600 ants). This method was selected as ants proved less likely to squirt formic acid (and therefore lose weight) when removed using an aspirator than by forceps. The forty ants were collected from each tree in quick succession, irrespective of the tasks they were performing, but ants not travelling in a clear direction were avoided. For collected individuals, we recorded whether the ant was carrying anything (needles or leaves or arthropod prey) and its liquid feeding status. Liquid feeding status was assessed by examining the gaster of individuals and allocating them to the following classes: (1) full: arthrodial membrane stretched, such that the length of arthrodial membrane visible along the mid-line of the gaster was at least half of that of the sclerites; (2) half full: some stretching of the arthrodial membrane, but with the visible length less than half that of the sclerites; (3) empty: no visible stretching of the arthrodial membrane.

Ants were killed or subdued using ether in the field and were later frozen for three days at −20 °C. They were weighed in groups of twenty ants travelling in the same direction on the same tree (tree was the replicate measure in this study and ants were weighed in bulk to improve accuracy). Any needles or prey items were removed before weighing. The total mass of ants was divided by twenty to obtain a mean mass per ant and the mean mass of the "Up" ants was subtracted from that of the "Down" ants to determine the average mass gain of ants on each tree.

## Study 2: Are foraging loads of individuals affected by resource quality and stand age?

As described above, twelve spruce-dominated stands (4 mature, 4 middle-aged and 4 clear-cut) were used for this component of the study. To compare the mass of honeydew collected by individual ants in different stand ages, we collected ants moving up and down spruce trees between 9:00 and 17:00 on fine days in July 2008 using the methods described above. Temperatures varied from 11.3 to 27.9 °C during these surveys. Ten spruce trees (basal diameter range: 0.6–61 cm) were selected at each site between 0.5 m and 35 m from a central nest (this distance range was necessary to include sufficient trees on clear-cuts). Central

nests were greater than 60 m from any edge with stands of a different age category. Trees with high *F. aquilonia* activity were preferred in order to reduce collection times. Using an aspirator, we collected five ants moving up and five ants moving down each of the ten trees at four sites belonging to each of the three stand ages for a total of 1,200 ants. We did not specifically select ants with laden and non-laden gasters because we aimed to compare harvesting loads per ant. Ants were placed in a cold box (approximately 5 °C) in the field and were later frozen for 3 days at −20 °C to ensure that they were killed. Ants were weighed individually (mass range: 2–21 mg) in the lab using a Mettler AE166 balance (sensitive to ± 0.1 mg) and maximum head width (range: 1.03–2.06 mm) was measured using an eyepiece micrometer on a Leica MS5 microscope. Any needles or prey items were removed before weighing.

## Statistical analyses

For study 1, a paired $t$-test on JMP (*SAS-Institute, 2007*) was used to compare the mean mass of ants walking up and down per tree. We used a general linear mixed model to test the effects of the fixed predictors temperature, tree species, diameter, $\log_{10}$ distance and their interactions and the random predictor nest, on: (1) the mean mass gain per ant; and (2) the exponential-transformed percentage of individuals carrying discernible liquid loads. Preliminary analyses, where tree height + distance was used instead of distance gave similar results to the final analyses and, as we were unsure how high up the tree aphids were located, we used distance to the tree base in all analyses. Distance was $\log_{10}$-transformed to improve model fit. We tested all possible models on MuMin (*Barton, 2011*) in R (*R Development Core Team, 2013*) and, because there was no clear best model, we performed model averaging of models within 2 AICc of the best model to determine the relative importance of variables in the set of best models (*Burnham & Anderson, 2002*). We present the coefficients from the model-averaged model with shrinkage and the importance of each variable among the set of best models. A $z$-test was used to compare the slopes of the ant mass (mean per tree)—distance from nest relationship for: (1) ants walking up and; (2) ants walking down a tree.

We modelled the effect of distance from a nest and tree diameter on harvesting rates at the colony level using the parameters from the model-averaged model predicting mean mass gain per ant from study 1 (described above) and the best model predicting ant activity per minute at the study site, using data from a previous study (*Gibb & Johansson, 2010*). The rate of ant activity was determined by recording the number of ants crossing a line 10 cm from the base of the tree in one minute. To illustrate responses across a distribution of diameters typical of mature stands, we present results for trees of basal diameter 10, 25 and 40 cm. We used set values of 22 °C for temperature (the mean value during surveys) and spruce for tree species (the most commonly occurring tree species) for the modelled data.

Finally, for data from study 2, a general linear mixed model with head width as a covariate and site as a (random) blocking factor, was used to test the effect of temperature, stand age, distance ($\log_{10}$-transformed), diameter and their interactions on the mass of ants walking down trees. We used only ants walking down trees because analyses for study 1 showed that distance-load relationships were significant only for these ants. We included head width to

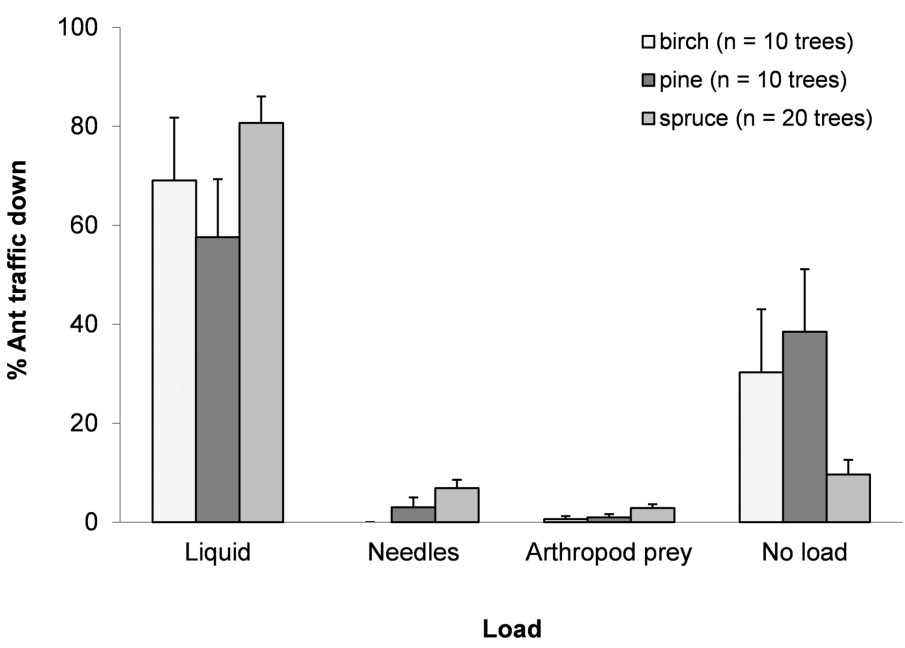

**Figure 1 Mean ± SE percentage of total trips for each resource collected.** Ants collected honeydew, needles, arthropod prey or nothing apparent for birch, pine and spruce. $N = 1,598$ trips.

account for size-related load capacities of ants (*Wright, Bonser & Chukwu, 2000*). We again tested all possible models on MuMin and performed model averaging of models within 2 AICc of the best model to determine the relative importance of variables in the set of best models. We present the coefficients from the model-averaged model with shrinkage. We report both marginal (fixed effects; $R^2_{GLMM(m)}$) and conditional (fixed + random effects; $R^2_{GLMM(c)}$) $R^2$ values (*Nakagawa & Schielzeth, 2013*), calculated using the package MuMIn. We also used ANOVA on JMP to test the effect of stand age on microsite temperature.

## RESULTS

### Overview of ant loads

On average, ants walking up a tree weighed significantly less than those walking down ($t_{(1,39)} = -9.766$, $p < 0.0001$), gaining $2.11 \pm 0.19$ mg (Mean ± SE) in mass, or approximately 33% of the average mass of an ant walking up the tree (up ants: $6.42 \pm 0.17$ mg; down ants: $8.53 \pm 0.27$ mg). Observations of the loads carried by ants suggested that most ants walking down a tree were carrying a discernible honeydew load (full or half full) (70.4 ± 4.9%). Although the mean weight gain for ants was 33%, not all ants carried honeydew loads, indicating that ants with loads carried around 47% of their body weight. Among the collected ants, $8.0 \pm 4.0\%$ carried nest material from trees, i.e., needles (birch leaves were never taken) and $3.5 \pm 1.7\%$ carried arthropod prey (mainly aphids and spiders) (Fig. 1).

### Do ants adjust their foraging based on resource quality and distance?

Analysis of the effects of quality (tree species and basal diameter) and distance of the tree from a nest on mass gain revealed a reasonable fit, with little contribution of random effects

**Table 1 Estimates and importance ($\Sigma w_i$) from model-averaged models for mass gain and liquid loads at the site level.** Models tested the effects of tree species, temperature, distance, basal diameter and their interactions on the mean mass gain and the percentage of individuals bearing liquid loads in their gasters. Random effects not shown.

| Source | Mass gain | | | % Liquid bearers | | |
|---|---|---|---|---|---|---|
| | Estimate | (SE) | $\Sigma w_i$ | Estimate | (SE) | $\Sigma w_i$ |
| Intercept | 3.00 | (1.13) | | 1.64 | (0.42) | |
| Tree species (pine) | 0.05 | (0.26) | 0.25 | | | |
| Tree species (spruce) | −0.13 | (0.30) | | | | |
| Temperature | −0.03 | (0.05) | 0.30 | 0.01 | (0.02) | 0.31 |
| $Log_{10}$ (distance) | 0.52 | (0.18) | 1.00 | 0.07 | (0.06) | 0.75 |
| Diameter | −0.05 | (0.02) | 1.00 | | | |

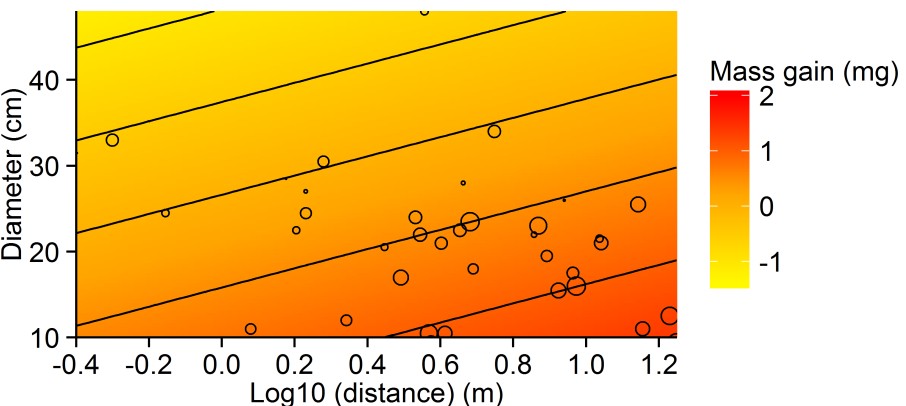

**Figure 2 Contour plot showing the relationship between tree diameter, $log_{10}$ distance and mean mass gain per ant for the small-scale study.** Circles represent values for mass gain, ranging from −0.74 mg (smallest circles) to 2.11 mg (largest circles). Contour bin width is 0.5 mg.

($R^2_{GLMM(m)} = 0.41$, $R^2_{GLMM(c)} = 0.41$) (Table 1). Mass gain was negatively, but weakly related to tree diameter, suggesting individual ants gained less on larger trees (Fig. 2). Mass gain was positively related to distance to nest, with ants travelling farther carrying heavier loads (Fig. 2). Tree species was of low importance in predicting mass gain. The model showed weaker predictive power for the percentage of workers carrying observable honeydew loads, with the random factor 'nest' contributing most to model fit ($R^2_{GLMM(m)} = 0.11$, $R^2_{GLMM(c)} = 0.49$). The percentage of liquid bearers responded only weakly to distance, while tree species and diameter did not appear in any of the best models.

The relationship between distance and mean mass of ants was significant only for ants walking down trees. Slopes for the relationship between distance and mass were significantly different for ants walking up and down trees ($Z = 3.37$, $p < 0.0005$) (Fig. 3). This suggests that the distance a worker travels was not determined by its size, but that ants walking farther acquired a larger load.

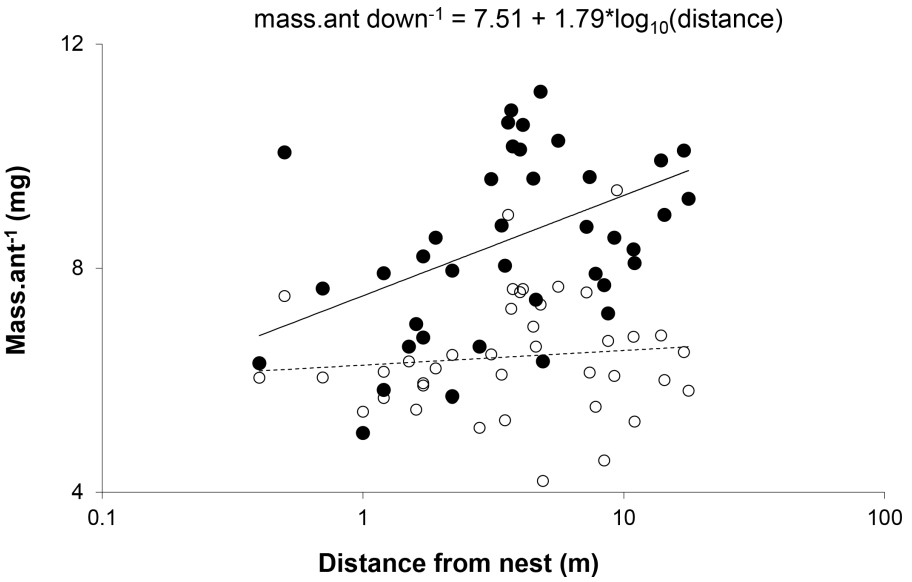

**Figure 3** **Mean mass per ant walking down (●) and up (○) each of the trees plotted against distance from the nest.** The slope for the relationship between weight and distance travelled was significant for ants walking down trees ($F_{(1,38)} = 8.520$, $p = 0.0059$, $R^2 = 0.18$), but not those walking up ($F_{(1,38)} = 0.403$, $p = 0.5296$, $R^2 = 0.01$). $X$-scale is logarithmic.

The model incorporating the mass of honeydew harvested per ant (parameters in Table 1) and ant activity per tree suggests that the effect of distance from the colony on the mass gained by the colony at each tree depends on tree basal diameter (Fig. 4). Trees with larger diameters (40 cm) made relatively consistent contributions to mass gain, independent of distance (within 20 m from a colony), while the contribution of honeydew from smaller trees (diameter = 10 cm) decreased with increasing distance from a colony. This is because mass gain increased, while activity decreased with distance and activity increased, while mass gain decreased with diameter.

## Are foraging loads of individuals affected by resource quality and stand age?

The model-averaged model testing the effects of head width, temperature, stand age, tree diameter, distance and their interactions on mass of ants walking down trees was a good fit to the data ($R^2_{GLMM(m)} = 0.62$, $R^2_{GLMM(c)} = 0.66$). Interactions between stand age and distance did not appear in the model. Stand age*diameter and distance*diameter interactions were included in the model, but had low importance (Table 2). The covariate 'head width' was an important predictor of worker mass, as expected. Consistent with survey 1, distance was positively related to the mass of workers walking down trees. In contrast to study 1, tree basal diameter had a positive effect on the mass of workers climbing down trees. However, the importance of diameter was low in the model for study 2, suggesting a weak relationship. Microsite temperatures measured during the surveys were not significantly higher at clear-cuts (mean ± SE: 19.2 ± 2.1 °C) than mature (16.2 ± 0.5 °C) or middle-aged stands (18.5 ± 2.0 °C) (ANOVA: $F_{(2,9)} = 0.87$, $p = 0.451$).

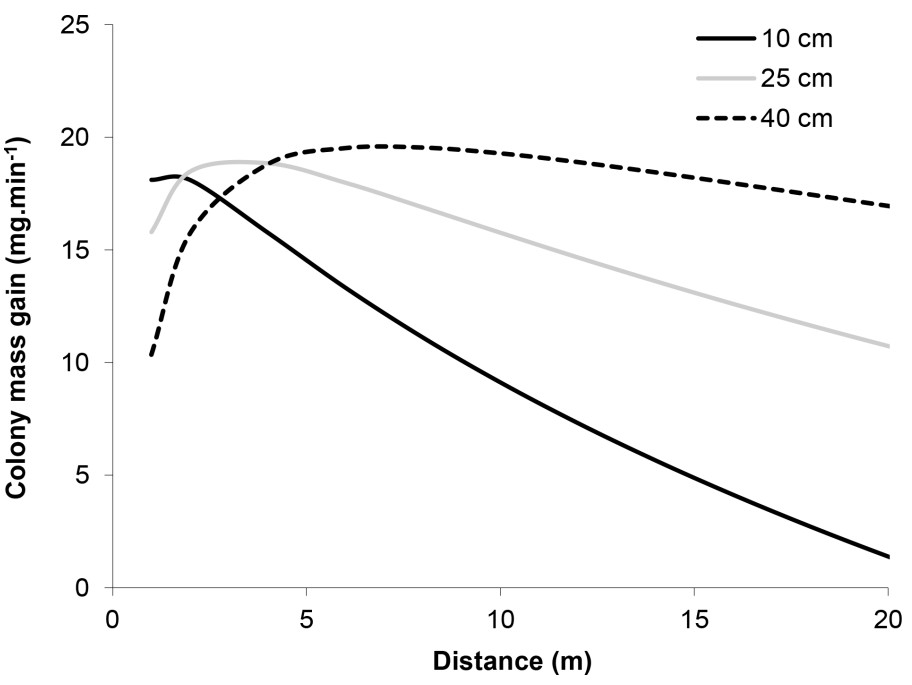

**Figure 4** Model of the relationship between the mass of honeydew gained by the colony per minute and the distance of the tree from the nest at three different tree basal diameters (BD = 10, 25 and 40 cm). The estimate was calculated by multiplying equations for mass gain per ant (Mass gain.ant$^{-1}$ = 3.82 − 0.05*Temperature + 1.11*Log$_{10}$ (distance)−0.06*Diameter) by ant activity per minute (Activity.min$^{-1}$ = −16.83 + 1.02*Temperature −2.18∗ Log$_{10}$ (distance) + 0.12*Diameter). Calculations were made for spruce trees in mature forests at 22 °C.

**Table 2** Estimates and importance ($\Sigma w_i$) from model-averaged models for worker mass across stands. Models tested the effects of stand age, temperature, distance, basal diameter and their interactions on the mass of workers climbing down trees. Random effects are not shown.

| Source | | Estimate | (SE) | $\Sigma w_i$ |
|---|---|---|---|---|
| (Intercept) | | −1.02E−02 | (7.78E−04) | |
| Head width | | 1.17E−02 | (4.22E−04) | 1.00 |
| Temperature | | −2.65E−06 | (1.45E−05) | 0.13 |
| Stand age | 80–100 yrs | 1.41E−04 | (4.25E−04) | 0.14 |
| | 30–40 yrs | 5.88E−05 | (2.72E−04) | |
| Log$_{10}$(distance) | | 3.29E−04 | (1.05E−04) | 1.00 |
| Diameter | | 1.96E−05 | (3.34E−05) | 0.75 |
| Diameter*stand age | 80–100 yrs | −1.27E−05 | (3.45E−05) | 0.14 |
| | 30–40 yrs | −8.05E−06 | (2.59E−05) | |
| Diameter*Log$_{10}$(distance) | | 1.02E−06 | (3.91E−06) | 0.16 |

## DISCUSSION

### Resource harvesting

A high percentage of ants walking down trees (approximately 70%) appeared to be carrying honeydew in their gasters, suggesting that this was their main task in climbing trees. This is as expected as honeydew constitutes 78–92% of the diet of wood ants (*Domisch et al., 2009*). Of the 30% of ants for which no discernible stretching of the gaster was observed, it is possible that many were involved in other activities, such as guarding resources, or that they had collected much smaller volumes of honeydew. Smaller percentages of ants returning from trees collected arthropod prey and nest material. While it might appear opportune to collect arthropod prey if encountered in the canopy, the collection of nest material from such a height and in trees up to 14 m from the nest was unexpected. Ants collecting needles from pine or spruce canopies in older forests travel considerably farther than would appear necessary, given that needles are abundant on the forest floor. A possible explanation is that needle quality is better if needles are removed directly from the tree, perhaps because micro-organism activity is lower on such needles. The lower carbon to nitrogen ratio of needles found on ant mounds, relative to those found on the forest floor (*Kilpeläinen et al., 2007*), suggests that harvesting of needles from the canopy may be common practice for *F. aquilonia*. However, further sampling is required to properly address this supposition.

### Do ants adjust their foraging based on distance?

The distance travelled positively affected the load collected by ants. Given that modern formulations of central place foraging theory do not support a positive load-distance relationship (*Olsson, Brown & Helf, 2008*), a likely explanation for the observed positive load-distance relationship is that high activity of ants on trees near mounds might lead to crowding and faster turnover of workers feeding from aphids, resulting in smaller loads as a consequence of reduced time spent harvesting due to physical interference (defence) or overexploitation of the resource (*Sundström, 1993*; *Wright, Bonser & Chukwu, 2000*; *Grüter et al., 2012*). Such a density-dependent response could also explain our finding that, although ants were more active on large-diameter trees, they harvested less honeydew. This is further supported by the appearance of distance as an important predictor of the percentage of individuals with liquid loads among the best models (Table 1), indicating more workers engaged in tasks other than harvesting at trees closer to the nest. Engagement in other tasks (e.g., *Novgorodova, 2015*) might also explain the slightly lower colony mass gain for trees within a few metres of the nest (Fig. 4).

In contrast to previous studies, which showed that larger ants travel greater distances to forage because they are able to carry a greater load relative to their body weight (e.g., *Herbers & Cunningham, 1983*; *Rosengren & Sundström, 1987*; *McIver & Loomis, 1993*; *Wright, Bonser & Chukwu, 2000*), size did not determine distance travelled: we found a significant positive relationship between mass and distance for ants walking down trees but not for ants walking up trees. This suggests that, within the range of distances examined in this study, there is no distinct division of labour depending on worker size, but that individual workers that have travelled further collect more honeydew, possibly due to the density effects discussed above.

## Do ants adjust their foraging based on resource quality?

Animals are expected to allocate more effort to harvesting resources of higher quality (optimal foraging theory, *MacArthur & Pianka, 1966*). Individual honeydew loads did not differ among tree species, despite differences in sugar composition (T Johansson, 2008, unpublished data). In contrast, ants in mature forest responded to trees differing in diameter as though they differed in quality, harvesting more from small-diameter trees. Differences between small- and large-diameter trees in the mature forest may reflect differences in the quality of honeydew as a result of changes in tree defence against aphid herbivory with age. Vigorous and/or fast growing plants usually have fewer secondary metabolites and are therefore often preferred by herbivores, including aphids (*Whitham, 1978*; *Price, Roininen & Tahvanainen, 1987*; *Price, 1991*). However, *Johansson & Gibb (2012)* showed that spruce trees in mature forest, which tend to be of larger diameter, have a greater concentration of some sugars attractive to ants than young trees regenerating in clear-cuts. This suggests that sugar quality may actually be higher in large-diameter trees, so resource quality is unlikely to be a driver of this difference.

Despite the smaller individual loads harvested from larger trees, colonies allocated more workers to large trees. Models showed that the net result of opposing allocation of worker activity and individual loads was that, close to the nest, a similar mass of honeydew was harvested from small and large trees (Fig. 4). However, further from the nest, larger honeydew loads from smaller trees did not compensate for the greater activity on larger trees. Thus, the colony-level mass gain from large trees remained relatively constant with distance, while the mass gain from smaller trees declined rapidly.

At the colony level, a decline in selection of smaller trees with distance might occur if small trees act as small resource patches. This is in agreement with previous studies that have shown greater recruitment to better quality (or larger) resource patches (*Taylor, 1977*; *Nonacs & Dill, 1991*). However, another possible explanation for the rapid decline in use of small trees with distance may be related to the limitations of ant navigation, which, for *Formica spp.*, is largely dependent on memorising the location of landmarks (*Graham & Collett, 2002*; *Fukushi & Wehner, 2004*). Larger trees may therefore present a clearer image for visual memory, although the panorama, rather than individual features, may be critical for navigation (*Collett, 2009*). Alternatively, foraging paths can be costly to maintain in complex habitats (*Shepherd, 1982*) and the low rate of return of workers from small trees may be insufficient to sustain recruitment (*Pinter-Wollman et al., 2013*), so resources available from smaller trees may fall below the threshold under which trail maintenance is efficient.

## Is foraging load affected by stand age?

Although the total quantity of honeydew harvested per hectare differs among stands of different ages (*Gibb & Johansson, 2010*), we detected no change in the foraging responses of individual ants. While the interaction between diameter and stand age appeared amongst the best models for mass gain, this relationship was of low importance. The basic rules that determine the behaviour of individuals were minimally altered by anthropogenic disturbance alone. However tree diameter and distance, which vary with stand age, were important predictors in the set of best models. This suggests that other factors that vary with

stand age, such as time since disturbance, competition or predation are not important in determining foraging load. Previous studies suggest foraging behaviours of ants and other species are often affected by anthropogenic disturbances (*Mahan & Yahner, 1999*; *Goverde et al., 2002*; *Sorvari & Hakkarainen, 2004*). However, changes in behaviour may reflect attempts by individuals to maximise foraging loads in the new environment, so may not alter this fundamental relationship.

## Conclusions

Our findings suggest that most *F. aquilonia* workers observed on trees are engaged in honey-dew harvesting. Distance had a clear positive effect on worker loads. However, for both distance and diameter, worker activity declined as loads increased, indicating a role for crowding in reducing colony-level efficiency (*Dreisig, 1988*; *Grüter et al., 2012*). Load-distance relationships were consistent across stands of different ages, suggesting no effect of stand age on this fundamental response, despite effects of stand age on activity and honeydew quality (*Gibb & Johansson, 2010*; *Johansson & Gibb, 2012*). This is in contrast with findings suggesting effects of anthropogenic disturbances on a range of behavioural responses. However, changes in behaviour may often occur to improve foraging loads or nesting success. Behavioural changes in responses to disturbance may thus tend to be consistent with maximising foraging loads if they are within the evolutionary experience of a species.

# ACKNOWLEDGEMENTS

We would like to thank Åke Nordström and Ylva Nordström for invaluable assistance in the field.

## Funding

Funding for this project was provided by Formas (the Swedish Research Council 2005-3012-3674-25) and Lilla Fonden (the Swedish University of Agricultural Sciences). The funders had no role in study design, data collection and analysis, decision to publish, or preparation of the manuscript.

## Grant Disclosures

The following grant information was disclosed by the authors:
the Swedish Research Council: 2005-3012-3674-25.
the Swedish University of Agricultural Sciences.

## Competing Interests

The authors declare there are no competing interests.

## Author Contributions

- Heloise Gibb conceived and designed the experiments, performed the experiments, analyzed the data, wrote the paper, prepared figures and/or tables.
- Jon Andersson and Therese Johansson performed the experiments, reviewed drafts of the paper.
### Field Study Permissions

The following information was supplied relating to field study approvals (i.e., approving body and any reference numbers):

This study complies with the current laws of Sweden. The forestry companies Holmen Skog AB, Sveaskog AB, SCA and Scaninge gave permission to use their land.

### Data Availability

The raw data is available as Data S1.

### Supplemental Information

Supplemental information for this article can be found online at http://dx.doi.org/10.7717/peerj.2049#supplemental-information.

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
