# Peer review of "Foraging loads of red wood ants: Formica aquilonia (Hymenoptera: Formicidae) in relation to tree characteristics and stand age"

_PeerJ, doi:10.7717/peerj.2049_

## Round 0.1 · original submission · Minor Revisions

· Academic Editor

Minor Revisions

Overview

This study examined the foraging behavior of wood ants collecting honeydew from aphids living on trees in managed forests. In one study of 10 nests in a mature forest, the honeydew load size increased with distance from the nest, decreased with tree size, and was not obviously affected by tree species. Ant size did not vary with distance, and the proportion of ants carrying honeydew showed little, if any, relationship to tree distance, size or species. When load size data were combined with ant activity data collected in a previous year, it appeared that on trees close to the nest, colonies harvested more honeydew from smaller trees than from larger trees whereas for trees far from the nest, more honeydew was harvested from larger trees. In a second study carried out in multiple stands of different ages, the positive effect of distance on load size remained along with a possible effect of tree size but there was no effect of stand age that was not better accounted for with tree size. The authors interpreted their data within an optimal foraging framework while considering alternative explanations.

We received three reviews of this manuscript as a result of the timing of reviewer acceptance of the PeerJ invitations. This was a useful occurrence because each reviewer provided distinct suggestions for improvement of the manuscript, while agreeing that it was a valid contribution to the field. I agree with the positive assessment but feel that the manuscript could be substantially improved by clarification of the framework of the study, confirming the validity of the statistical approach, and clarifying some possible errors and inconsistencies as noted by the reviewers and my detailed comments below. While the suggested changes are numerous, I consider them to be minor revisions. Assuming that no additional issues arise and that my comments and those of the reviewers are satisfactorily addressed, I hope that it will not be necessary to send the manuscript out for additional review.

Note that you can respond to my specific comments as you would to any reviewer. After consideration, i.e. either making the suggested changes or thoughtful rejection are acceptable.

Major issues
1) Framework. The scientific structure of the study is ambiguous. The Abstract and first part of the Introduction imply that this is a test of optimal foraging theory. However, the study objectives do not refer to theory but imply an examination of relationships among variables. The Discussion includes brief reference to theoretical expectations but no rigorous discussion about whether the underlying hypotheses are supported or rejected.

A critical issue in a test of theory is to assess whether the predictions actually apply to the test situation. With regard to the positive relationship between load size and distance, this prediction is only valid when the rate of resource gain decreases as load increases. No evidence is presented to indicate that this assumption is even valid for ants gathering honeydew. This decreasing gain rate seems much more critical than the positive relationship between distance and energy expenditure as implied on L52.

Furthermore, Reviewer 2 points out that this classic prediction from central place foraging theory is theoretically valid only in a between-environment context. I was among the earliest researchers to test the Orians and Pearson load size predictions but, not having worked on these questions for some time, was unaware of Olsson et al.'s (2008) interesting article. It made me wonder whether my research in the 1980s had found solid qualitative support for an invalid prediction and whether similar concerns applied to your situation. After thinking about this for a while, I suspect that the prediction of no change in load size at different distances would apply only when the animal randomly encounters patches at different distances, paralleling marginal value theorem assumptions about between-patch travel costs. Many central place foragers that exploit large patches return repeatedly to the best available patch and thus distances are not randomly encountered. This probably makes the 'between-environment' predictions apply within the same home range as the location of the best patch changes. It is possible that these considerations would be valid for your system also, but the competition among individuals and optimization at the colony level add additional complications that need to be thought through.

The prediction that larger loads will be taken from better patches also has some critical assumptions and would not be valid for many definitions of 'better' or 'high quality'. Your argument on L283ff incorrectly equates effort and load size and greatly simplifies the prediction. Again competition or interference among individuals within the colony may be an important consideration.

It is important to address the complexities mentioned if you wish to maintain your presentation as a test of predictions of optimal foraging theory. Alternatively, you could revise your framework to present your study as an examination of factors influencing load size and your other dependent variables, explaining the importance of investigating these variables and simply explaining the potential relevance of your selected independent variables based on a combination of previous theory and empirical results, despite not being able to directly test the theory.

2) Statistics
Reviewer 2 raises a concern about mixing information theoretic and hypothesis testing approaches. My understanding is that such an approach is not recommended. There is ambiguity in what effects you indicate as supported, apparently considering some as supported as a result of model averaging, even though the probability is considerably greater than 0.05. Reviewer 1 raises additional concerns regarding the transformation of proportions, the model selection process and the implications of multiple testing. In addition, I believe that there may be a problem with models incorporating highly correlated variables such as stand age and tree size in your Study 2. Would it be desirable to first test stand age without tree size and then show that the effect can be better explained by tree size as a correlated variable? Given these concerns, you should ask a biostatistician to carefully review your statistical methods and inferences, revise or clarify your conclusions as necessary, and provide a more explicit statement in the Statistical Methods section regarding your criteria for accepting or rejecting the influence of specific variables and for concluding that their effects were strong or weak.

Minor issues
L1. In the title and elsewhere (e.g., L164), I question whether the term 'efficiency' is appropriate. I think of foraging efficiency as measured by energy (or some other resource) gained divided by energy expenditure. I don't see how load size can be considered a correlate of efficiency without any measure of cost.
L22 and Introduction. Energy maximization is an assumption of optimal foraging theory used to generate predictions, not a conclusion. Note also that the assumption is maximization of net rather than gross energy gain. As Reviewer 2 notes, more recent approaches also take mortality risk, nutrients and other currencies into account. You do not need to consider all these, but you do need to properly frame your objectives.
L25. You do not have the data to assess whether successional stage affects optimization. You can ask whether it affects the relationship between load size and distance, but you do not know what relationship is optimal. There could be lots of potential positive correlations that would be sub-optimal. Potentially, you could have asked whether the load size distance relationship is quantitatively similar in the different successional stages, but I suspect that the change in methods prevents you from doing so.
L30 and elsewhere. As Reviewer 3 notes, your studies depend on observation of relationships among variables, not manipulation of potential causal factors. Therefore, you should be cautious in inferring causal direction. Because your design is observational rather than experimental, it would be preferable to refer to the two investigations as Study 1 and Study 2 rather than Experiment 1 and 2 (although I do recognize that not all researchers use this strict definition of an experiment).
L34ff. In the Abstract, it is hard to relate your results to your goals. The findings related to the second objective are not mentioned and the third objective is mentioned only indirectly in the results section but then both are raised in the discussion section. Tree species is not mentioned. In addition, please check that effects and whether they are weak and strong are fully consistent with your statistical evidence and with a clear exposition of the pattern in the Results.
L46. Consider adding 'central place foraging' to keywords.
L78-81. Sentence is unclear. To indicate that stage increases competition, you must specify which stages are being compared and which are higher. For predation and resource quality, it would not take many more words to indicate the direction of the relationships rather than simply indicating a change.
L85-86. I think you mean that selection rather than evolution operates at the colony level. You should have a reference to support the assertion.
L88ff and elsewhere, e.g., L125ff. In defining your objectives, you refer to effects of resource quality, but quality is defined above as sucrose concentration and carbohydrate:protein ratio, which you did not measure. I think you would be better to be specific about tree species and size, which are the variables you actually measured. It would be very helpful to the reader to explain how these variables might be specifically related to quality as defined, not just that they are relevant variables. This point is also relevant to the Discussion.
L102. Revise the wording: your investigation was not a detailed study of the theory per se.
L114. If the concept is important, you should explain what territorially dominant ant species are for readers who do not work on ants.
L132. Is the range of times of day potentially important to someone attempting to replicate your research?
L139ff. Do you have any evidence about loss of load from this method and the one used in Study 2, e.g., releasing liquid in response to aspiration, ether or freezing?
L160. Is there any potential bias introduced by selecting trees with high ant activity? What was the distribution of tree sizes examined in relation to their availability? The reader does not have an idea of the range of values for any of your variables.
L185. Please provide an operational definition and units of ant activity per minute. In many areas of ecological research, activity is the proportion of time that an animal spends moving, which does not seem to be the definition in your case.
L193. Why did you switch methods between study 1 and 2? Measuring only ants coming down in study 2 prevents you from comparing load sizes with study 1 and requires assuming that the previous lack of an ant size-distance relationship still holds. If this was your plan, what was the reason for collecting ants going up? Note that the Results imply that you measured gain (L235), but this is not possible with the method as described.
L206. Can you affirm that no ants walking up carried honeydew? I am not sure that ants always main a consistent direction of movement, whether loaded or not.
L208. Rearrange sentence to avoid starting with a number.
L213. Do you know how high the ants climb in the trees? Is movement on the trunk and main branches part of distance traveled? Does this confound tree size with distance? (Potentially these are issues for Methods and/or Discussion.)
L217, 225, 298. 'Farther' rather than 'further' is preferred for physical distances.
L218. Shouldn't you specify that tree species had no detectable effect? Shouldn't the equation for load size used in Figure 4 be explicitly referred in the Results?
L224. It is not clear why the regressions for ant size on distance used ln transformation whereas you used log10 transformation in the models. Also, readers would probably find it easier to interpret the patterns if you provided arithmetic values for distance even though the scale is logarithmic.
L228. Is there a reason why you ignored the decrease in load size for 40 cm trees at distances < 2 m?
L231. This sentence is very hard to understand, in part because of placement of commas. Please revise, perhaps splitting into two sentences.
Figure 4. I don't understand the caption here. Why do you present the constant as a product and difference instead of a single number? For example, in the load size equation, why couldn't 2.87-0.03*18.9 be replaced with 2.30? Also, it seems awkward to mix log10 and ln as transformations of the same variable. Furthermore, the text (L232, 276, 295-6) refers to opposing effects of tree size on load and activity, but the coefficients are negative in both equations.
L236. I think it is normal to present main effects before interactions. The Results for Study 2 do not mention the effect of distance, the lack of effect of stand age, or the interpretation of the possible tree size effect.
L241. Included the units for temperature.
L257. Capital letters not needed for carbon and nitrogen.
L266 (and L83). In checking to see whether other ant studies use the term efficiency as you did, I noticed that Wright et al. 2000 carried out a very similar study to yours. Given that, it may be appropriate to cite their precedent in the Introduction, comparing the goals of your study to what they had published and to explicitly compare your findings to theirs in the Discussion. I did not expect to find such a similar study because of the small number of references you gave to their work and because their work was included in lists of multiple references rather than having their previous work compared to yours.
L272ff. The alternative hypothesis based on crowding needs clarification. The concepts of worker turnover and resource defense need to be explained to allow the reader to understand the processes involved. For example, where is the turnover occurring and what is being defended and by whom? Is there any evidence from other studies with a similar context to yours that aphid honeydew production can become a limiting resource for ants?
L331. I think it is an overstatement of your results to indicate that load-distance relationships were consistent across stands. You only show a general qualitative trend and the lack of a difference may be related to the power of your study, which you never discussed.

·

Basic reporting

Any areas not specifically referred to meet the standards of PeerJ, to my knowledge.

• “The article should include sufficient introduction and background to demonstrate how the work fits into the broader field of knowledge. Relevant prior literature should be appropriately referenced.”

The handling of literature was generally good, and gave sufficient details for an introduction to the topic. However, the lack of any reference to the large literature on leafcutter ant optimal foraging was very conspicuous by its absence. A lot of work has been carried out on leafcutter ants on effectively very similar questions to the ones posed here (the effect of distance, resource quality, and other tree attributes of load selection). Work by Martin Burd and Flavio Roces will be prominent in this literature, and more work is appearing all the time. Some selected citations might include (Roces 1990; Burd 1995; Burd 1996; Burd 2000), but these are not necessarily the most relevant works; just something to get the authors on the trail.

One idea which was missing to me for both the introduction and especially the discussion was the importance of “foraging for information”, and the role of information transfer in deciding how large a load to select (or how long to spend foraging). This idea was championed by Flavio Roces (see for example (Roces and Bollazzi 2009). I think these concepts have much relevance for this work.

• “Figures should be relevant to the content of the article, of sufficient resolution, and appropriately described and labeled.“

Figure 2 is somewhat hard to evaluate, as the dots are really rather small. It is a minor quibble, but could all the dots be made 50% larger?

• “All appropriate raw data has been made available in accordance with our Data Sharing policy.”

All data is presented, but a little effort could be made to explain the data – labels on the columns, and explanation of what the two tabs represent. This might seem trivial, but can mean the difference between usable and unusable raw data.

Minor comment – in lines 248-249, it would be worth citing (Novgorodova 2015), as this gives an interesting account of what else these ants might be getting up to. Makes for interesting reading in itself, as well.

Experimental design

• “The investigation must have been conducted rigorously and to a high technical standard.
• Methods should be described with sufficient information to be reproducible by another investigator.”

Were full ants tracked from the tree to the nest? The Formica rufa group ants are generally polydomous, and many of the nests are non-foraging (e.g. (Ellis and Robinson 2015), and other recent works by Ellis and Robinson). It may well be that the focal nest was not the nest to which the ants were returning, which would in turn invalidate many of the measurements taken. It is thus very important to note whether the authors are sure about where the ants are returning to.

Minor comment – in experiment 1, why were groups of ants weighed, not individual ants. I do not think this would have greatly changed the results, but it seems like information was needlessly wasted by grouping. It is now, of course, too late to rectify this, and this minor point does not invalidate the manuscript. Perhaps a note of explanation to prevent this irritating other readers?

Minor comment – stand age in in timber forests will affect many things, not just tree size. A key factor which is affected is tree density, with younger stands having a higher tree density but smaller tree sizes. This should probably be mentioned and discussed. To what extent are the results affected by tree density?

Related to this comment, the forests in which this study was conducted is a managed timber forest, correct? This means that there will be regular thinning as stands age. The time since last thinning might have important implications for aphid productivity, as just following a thinning there will be A) a lot less shading of the trees, and B) much more herbaceous growth in the understory. This is likely to affect aphid productivity, and thus the foraging decisions of the ants. Was this looked at or controlled in any way?

Minor comment – it is not clear why head width was included in this model (line 238) at all – is would obviously autocorrelate linearly with ant mass, especially as these ants do not show (to my knowledge) allometric head/mass relationships.

Validity of the findings

• “The data should be robust, statistically sound, and controlled.”

I am not qualified to evaluate the technical aspects of the modelling carried out in this work. The statistical analysis seems generally in order, but again I admit to only having a working knowledge of how such statistical tests should best be applied.

With that said as a caveat, I do have some comments on the statistical analysis:

1. Why was the data on percentage individuals carrying liquid load exponentially transformed (line 174)? I am given to understand that proportions (and percentages) are best logit-transformed before analysis in GLMMs (Warton and Hui 2011) but see (Shi et al. 2013).
2. The model selection method performed by the MuMin package seems to be an approach in which the saturated model is made, and non-significant terms sequentially removed. Some regard such methods as “fishing expeditions”, which are likely to turn up false positives (Forstmeier and Schielzeth 2011). Indeed, I prefer a more hypothesis-driven model selection procedure. However, the statistician-jury is still out on this.
3. There was no correction for multiple testing, but the final model contained several terms. Some researchers consider each separate term in a model to be a new test. The R-package Multcomp allows quick adjustment for multiple testing – might be worth looking at. This is especially important for the more borderline finding of this work; that “Mass gain was negatively related to tree diameter, suggesting individual ants gained less on larger trees”. Indeed, I doubt the biological significance of this finding, and would much prefer a less strongly worded approach to this in both the results and the conclusions.

• “The conclusions should be appropriately stated, should be connected to the original question investigated, and should be limited to those supported by the results. “

1. Related to the point above about the mass gain/diameter finding, what would the model in part 2 look like if we discounted this finding? This is an important point, as the mass gain/diameter finding is rather flimsy, and I do not think it bears the weight of further interpretation in a model.

Comments for the author

Apart from the comments above, all of which can hopefully be addressed rather easily, I found this manuscript to be of high quality, if not of breath-taking significance. The weakest part of the manuscript is the model, as it is not clear to me how much biological insight it really provides, seeing as it is based on one strong finding and one shaky one. I would like to see an alternative set of models where either both the high and low estimates of diameter effect are modelled, or the effect of dimeter is removed from the model. This would provide the more ‘conservative’ view of the results, and the conclusions that can be drawn from them.

All in all, I think this manuscript is worthy of publication following some revisions.

CITATIONS MENTIONED IN MY REVIEW

Burd M. 1995. Variable load size-ant size matching in leaf-cutting ants, Atta colombica (Hymenoptera: Formicidae). J. Insect Behav. 8:715–722.
Burd M. 1996. Foraging Performance by Atta colombica, a Leaf-Cutting Ant. Am. Nat. 148:597–612.
Burd M. 2000. Body size effects on locomotion and load carriage in the highly polymorphic leaf-cutting ants Atta colombica and Atta cephalotes. Behav. Ecol. 11:125–131.
Ellis S, Robinson EJH. 2015. The Role of Non-Foraging Nests in Polydomous Wood Ant Colonies. PLoS ONE 10:e0138321.
Forstmeier W, Schielzeth H. 2011. Cryptic multiple hypotheses testing in linear models: overestimated effect sizes and the winner’s curse. Behav. Ecol. Sociobiol. 65:47–55.
Novgorodova TA. 2015. Organization of honeydew collection by foragers of different species of ants (Hymenoptera: Formicidae): Effect of colony size and species specificity. Eur. J. Entomol. 112:688–697.
Roces F. 1990. Leaf-cutting ants cut fragment sizes in relation to the distance from the nest. Anim. Behav. 40:1181–1183.
Roces F, Bollazzi M. 2009. Information Transfer and the Organisation of Foraging in Grass- and Leaf-Cutting Ants. In: Food Exploitation by Social Insects. CRC Press. p. 261–275.
Shi PJ, Sand Hu HS, Xiao HJ. 2013. Logistic regression is a better method of analysis than linear regression of arcsine square root transformed proportional diapause data of Pieris melete (Lepidoptera: Pieridae). Fla. Entomol. 96:1183–1185.
Warton DI, Hui FKC. 2011. The arcsine is asinine: the analysis of proportions in ecology. Ecology 92:3–10.

·

Basic reporting

The whole paper, and in particular the portal paragraph, suffers from a rather shallow and far from up to date representation of foraging theory. Energy intake rate maximization has not been a corner stone of foraging theory for about 30 years (see for example any paper on the topic by McNamara and/or Houston, or on patch use theory by Brown). In addition, you claim that the positive relation between load size and distance is predicted by theory (in general), but in fact it is only in special cases. This relation might be the most misunderstood prediction in behavioural ecology. This prediction holds for so called between-environment comparisons. That is, if you compare one nest with its useful patches near, with another where useful patches are farther from the nest, the latter nest should take larger loads. However, if you compare two patches – one near and one far – from the same nest, the same load should be taken from both. This is obvious if you recall that the CPF (in the Orians & Pearson/Schoener version) is just a special case of the marginal value theorem by Charnov (1976). This was described by Olsson et al. (2008, Theor. Pop. Biol. 74: 22-33). In a within-environment comparison, they showed that load size is only going to be positively related to distance if better patches are farther from the nest. In other cases you should expect no relations or negative, if travelling carries direct costs. There might be additional cases that could generate a positive relation, such as efficiency maximizing, rather than long-term average rate maximizing. Thus, the primary expectation that you set out to test, does not follow from the theories you claim. I am aware that others have done the same mistake before you.

Somewhat related to this, is that you use the term “optimal” as if you a priori know that the ants follow optimal strategies (e.g. lines 102-103, 233, 313).

I found the discussions about the crowding effects (lines 272 and onwards) particularly interesting. This seems like a rather likely explanation for your results.

Experimental design

It appears that the experiments were well performed and relevant. However, it was hard to follow parts of it, and in particular to understand the designs of the two experiments in detail and the differences between them. Things such as size-density distribution of trees around each nest would have been useful to know. This lack of clarity unfortunately relates to all parts of the paper (introduction, methods, results, and to some extent discussion). I have therefore not been able to know for certain if your experiments are primarily within or between environment comparisons. If it would be possible to add variables for average tree density around the nests, and average patch quality that would make the results more interesting.

On the technical side I wonder why you have mixed statistical approaches. You both use hypothesis testing and an information based approach. Still you argue that you have clear hypotheses to test, and in a case like that I see little reason to do model averaging. This is exemplified by the statement on line 234: “The best model testing the effects of…”. This is a model arrived at by model averaging and cannot test any hypothesis in a formal sense.

Validity of the findings

The findings as such are probably valid, but not interpreted in relation to theory in an appropriate manner.

Comments for the author

I liked the experiments and the study at large. With some effort putting the paper into a context of contemporary understanding of foraging theory in general, and CPF in particular, I am sure this can be a valuable and worthwhile contribution.

Reviewer 3 ·

Basic reporting

This paper describes a test of optimal foraging theory in wood ants. The question is clearly defined, and so are the predictions. The abstract summarises the work well. The paper is superbly well-written throughout, with great figures and tables. It was a pleasure to read.

'Pass', subject to one issue detailed below:

Abstract
The final sentence of the results section of the abstract is confusing, it is not clear whether it intends to say that load is dependent on stand age (which is not clear from the data), or that diameter is dependent on stand age (which is obvious). Either way, it is ambiguous and needs improvement. If the sentence in fact intends to say that the relationship between diameter and load is dependent on stand age, then I think the authors are overstepping their data: the effect size that is being referred to here is so minuscule it's hard to believe this could possibly have any biological significance and it is very far from statistical significance, so while the interaction between stand age and diameter may be included in the best model I do not think that the authors can make this strong statement in the abstract.

Experimental design

'Pass', subject to further detail on the single point below:

Line 133. How were nests selected? Were they all mature? What was the minimum basal nest size?

Validity of the findings

'Pass', subject to correction of this important issue of the way the study is presented, i.e. being explicit it is an observational study and not a manipulation experiment.

Line 87 is poorly phrased. It states that the authors test whether individual workers adjust their loads with respect to resource distance and quality. However, this is not an individual-level manipulation experiment looking at how individual ants respond to distance and quality, and therefore we can know nothing about whether individual workers might adjust their loads. It could be that individuals that are naturally physiologically predisposed to carry heavy loads are also predisposed to travel longer distances before beginning to forage. The authors’ study is correlational so they cannot distinguish between these hypotheses. The data are still valid and interesting, but the authors need to phrase exactly what they are testing more clearly in the introduction.
Similarly in line 263 of the discussion, the authors state directionality in the relationship they observe. They posit that distance is affecting the load collected by the ants, but it is equally possible that the load capacity of the ants or their propensity to maximally fill themselves determines the distance they walk. While the former may be more likely than the latter, the authors' observations cannot distinguish between these and therefore they should change the sentence to "the distance travelled positively correlated with the load collected by ants".

Comments for the author

This paper describes a test of optimal foraging theory in wood ants. The question is clearly defined, and so are the predictions. The abstract summarises the work well. The introduction sets out the question and the background well. The results are clear, appropriately analysed and well-presented. The discussion is well-informed, thorough, and thought provoking. With the exception of the one point indicated above, speculations are clearly indicated as such. The paper is superbly well-written throughout. It was a pleasure to read and is a strong study which will make a valuable contribution to the literature.

---

## Round 0.2 · accepted · Accept

· Academic Editor

Accept

The manuscript has been carefully and thoroughly revised. I now consider it suitable for publication.